# Acute Effects of the Consumption of *Passiflora setacea* Juice on Metabolic Risk Factors and Gene Expression Profile in Humans

**DOI:** 10.3390/nu12041104

**Published:** 2020-04-16

**Authors:** Isabella de Araújo Esteves Duarte, Dragan Milenkovic, Tatiana Karla dos Santos Borges, Artur Jordão de Magalhães Rosa, Christine Morand, Livia de Lacerda de Oliveira, Ana Maria Costa

**Affiliations:** 1Postgraduate Program in Human Nutrition, College of Health Sciences, Campus Universitário Darcy Ribeiro, Universidade de Brasília, Brasília DF 70.910-900, Brazil; liviadelacerda@gmail.com; 2Unité de Nutrition Humaine, Université Clermont Auvergne, INRAE, UNH, F-63000 Clermont-Ferrand, France; dragan.milenkovic@inra.fr (D.M.); christine.morand@inra.fr (C.M.); 3Department of Internal Medicine, Division of Cardiovascular Medicine, School of Medicine, University of California Davis, Davis, CA 95616, USA; 4Laboratory of Cellular Immunology, Faculty of Medicine, University of Brasilia, Brasilia DF 70.910-900, Brazil; tatianakarlab@gmail.com; 5Laboratory of Food Science, Embrapa Cerrados, Planaltina DF 73.310-970, Brazil; artur.rosa@embrapa.br (A.J.d.M.R.); ana-maria.costa@embrapa.br (A.M.C.)

**Keywords:** *Passiflora setacea*, bioactive compounds, phenolic compounds, cardiovascular diseases, nutrigenomics, gene expression, immune system, cytokines, insulin, HDL

## Abstract

Background: *Passiflora setacea* (PS) is a passionfruit variety of the Brazilian savannah and is a rich source of plant food bioactives with potential anti-inflammatory activity. This study aimed to investigate the effect of an acute intake of PS juice upon inflammation, metabolic parameters, and gene expression on circulating immune cells in humans. Methods: Overweight male volunteers (*n* = 12) were enrolled in two double-blind placebo-controlled studies. Blood samples were collected from fasting volunteers 3 h after the consumption of 250 mL of PS juice or placebo (PB). Metabolic parameters (insulin, glucose, total cholesterol, high-density lipoprotein (LDL), high-density lipoprotein (HDL), and total triglycerides) and circulating cytokines were evaluated (study 1). Peripheral blood mononuclear cell (PBMC) from the same subjects were isolated and RNA was extracted for transcriptomic analyses using microarrays (study 2). Results: Insulin and homeostatic model assessment for insulin resistance (HOMA-IR) levels decreased statistically after the PS juice intake, whereas HDL level increased significantly. Interleukin (IL)-17A level increased after placebo consumption, whereas its level remained unchanged after PS juice consumption. Nutrigenomic analyses revealed 1327 differentially expressed genes after PS consumption, with modulated genes involved in processes such as inflammation, cell adhesion, or cytokine–cytokine receptor. Conclusion: Taken together, these clinical results support the hypothesis that PS consumption may help the prevention of cardiometabolic diseases.

## 1. Introduction

According to the World Health Organization (WHO), noncommunicable diseases (NCDs) are responsible for 71% of deaths worldwide, leading to the death of 15 million people aged between 30 and 69 years old. The most prevalent diseases are cardiovascular diseases, followed by cancers, respiratory diseases, and diabetes [1]. At the same time, the total number of people suffering from depression or other common mental disorders such as anxiety was estimated as exceeding 300 million people in 2015. These disorders are the biggest contributors to global disability and represent an important cost burden [2]. Therefore, stressful lifestyle markers such as emotional stress, an unhealthy diet (high in sugar, sodium, red meat, and trans fatty acids, but low intake of fruits and vegetables), overweight [3], and poor physical activity [4] increase the incidence of cardiovascular diseases (CVD) and NCDs. These lifestyle risk factors promote high blood pressure, hyperglycemia, hyperinsulinemia, hypertension, hyperlipidemia [1,5], obesity [1], high inflammatory cytokine production [6], and pro-atherogenic gene profile [7], and are associated with chronic low-grade inflammation and vascular inflammation [8].

A higher intake of fruit and vegetables is associated with a lower risk of all causes of mortality, particularly inflammation-related diseases [9]. Plant-based foods are sources of a variety of bioactive compounds (BC) such as terpenoids (carotenoids, essential oil components, phytosterols), polyphenols (flavonoids and non-flavonoids compounds) [10], sulfur compounds (glucosinolates and ally sulfinates), alkaloids [11], and polyamines [12], whose level of total intake is connected with the protection from chronic diseases, including cardiovascular diseases, cancers, and neurodegenerative diseases [11,12]. Several beneficial effects have been related to the consumption of these compounds such as antioxidant and anti-inflammatory activities [12,13]. These beneficial effects derive from the reported capacity of some BC to modulate cell signaling and consequently the expression of key genes [14]. The species of *Passiflora* genus have been studied due to their sedative, anxiolytic, anti-inflammatory, antioxidant, and anti-carcinogenic effects [15,16]. *Passiflora setacea* D.C (PS) is a wild passionfruit species of the Brazilian savannah, popularly known as “maracujá do sono” (“sleep passionfruit”). The consumption of these fruits has been traditionally associated with sleep modulation [17]. PS pulp and seeds have recently been identified as rich in BC, particularly in C-glycosides of flavonoids [18], and also homoorientin, vitexin, isovitexin, and orientin at higher contents than those found in *Passiflora edulis*, açaí (*Eurydema oleracea*), and orange juice [19]. They have also revealed antioxidant and antimicrobial properties in vitro [20,21]. These effects are potentially due to the presence of vitamin E and BC such as terpenoids, polyamines, and polyphenols, especially orientin, isoorientin, vitexin, and isovitexin [17,18,19,21]. These compounds have been reported to exert antioxidant, anti-inflammatory, vascular, neuroprotective, anxiolytic, and antidepressant-like effects [22,23,24,25,26]. Plant-based diets are recognized for their beneficial effects on the modulation of intermediate risk factors for inflammation-based disorders [27,28], and fruits constitute major contributors to these effects [29]. However, clinical studies focusing on the health properties of fruits of the Brazilian savannah, as well as on the potential underlying molecular mechanisms, including the modulation of expression of genes in humans, remain scarce. Therefore, the aim of our study was to evaluate the effect of acute consumption of PS pulp on metabolic and inflammatory biomarkers in overweight male volunteers, as well as to assess the impact on global gene expression profile in peripheral blood mononuclear cell (PBMC) by using microarray analyses.

## 2. Materials and Methods

### 2.1. Processing and Characteristics of Passiflora setacea Juice

The fruit used in this study was the PS, the BRS *Pérola do Cerrado* (BRS Pearl of the Brazilian savannah), which is cultivated at the experimental field of Embrapa Cerrados, Brasilia, Brazil, affiliated with the Brazilian Ministry of Agriculture. This study is part of a larger program called the Passitec Network, developed to improve fruit size and production. PS plants were cultivated in a vertical espalier system and the ripe fruits were harvested at their full maturity level during the rainy season, corresponding to the stage where the phenolics compounds were in their highest concentrations [19].

The pulps used in this experiment were prepared all at once and were aliquoted, thus allowing us to use the same batch of pulp throughout the study. The pulps were removed from the fruits and blended for 30 s to separate the seeds from the pulps by sieving. After that, they were aliquoted into portions of 150 g and placed into plastic bags, hermetically sealed, and stored at −80 °C. The batch of pulp used in this study contained 2.75 g/100 g in fresh weight (FW) of carbohydrates, 10.1 mg/100 g FW of vitamin C, 55.4 mg/100 g of proanthocyanidins, 86 mg gallic acid equivalents (GAE)/100 g FW of total phenolics, and 3.02 mg quercetine equivalents (QE)/100 g FW of total flavonoids in which we have results for the four main flavone C-glycosides (1.07 mg/g dry weight (DW) of orientin, 0.99 mg/g DW of isoorientin, 0.84 mg/g DW of vitexin, 1.13 mg/g DW of isovitexin) and for the flavanone glycoside (0.14 mg g^−1^ FW of hesperetin equivalent) [19]. The isocaloric placebo drink (PB) was obtained by mixing 100 mL of a passionfruit-flavored isotonic drink of the brand Gatorade with 150 mL of water to achieve the same final volume and sugar content of the PS juice (Appendix A).

### 2.2. Subjects and Study Design

Male volunteers (*n* = 12) were recruited by interviews after advertisements were published in the media (newspaper, website, etc.) from February to June 2015. Men, ranging from 40 to 64 years old, who were overweight or slightly obese (based on body mass index (BMI) between 25 and 31 kg/m^2^ or waist circumference >94 cm), non-smokers, and engaged in a low to moderate level (<5 h/week) of physical activity were eligible for inclusion. The exclusion criteria included a medical history of cancer or severe metabolic diseases, special dietary habits (e.g., vegetarians and vegans), use of dietary supplementation 2 months prior to the experiment (vitamin C, multivitamin, antioxidant capsules, etc.), chronic medication (anti-hypertensives, anti-hyperglycemic, anti-cholesterol, anti-depressants, anxiolytics, etc.), acute treatments 15 days prior to the experiment (anti-inflammatory drugs, antibiotics, etc.), and acute treatments 2 days prior to the experiment (inflammatory pain relievers such as aspirin, acetaminophen, etc.). A physical evaluation was performed to obtain measurements of weight, BMI, waist circumference, and percentage of body weight by applying the seven skinfold sites Jackson–Pollock method [30].

The study was performed in two phases. In both phases, the volunteers were asked to consume a “white meal”, which is a meal without foods rich in BC (vegetables, fruits, cocoa, and plant-based drinks) the day before the experiment (Appendix A). Seventy-two hours before the experiment, volunteers were asked not to consume alcohol or perform any kind of intense physical activity such as cycling and running. Study phase 1 aimed to set a controlled environment in which all volunteers would be offered the same food menu at the same time. For this, the volunteers (*n* = 12) were hosted for 2 days in a hotel. At day 1, blood samples were collected at fasting (T0) and 3 h after (T3) the consumption of 250 mL of placebo drink (PB). Similarly, on day 2, blood samples were collected at fasting and 3 h after the consumption of 250 mL of PS juice. Blood sampling and further biochemical analyses were performed by the Sabin clinical analysis laboratory, Brasilia. The results of the data obtained in study phase 1 are reported in Section 3.2 and Section 3.3.

After the first intervention, the study phase 2 aimed to investigate the effect of the consumption of PS juice on gene expression in the volunteers. The same volunteers were asked to consume the same “white meal” as in study phase 1 (Appendix A). The volunteers were invited to participate in a randomized crossover trial in which they had to acutely consume the same two beverages (250 mL PB or PS) in an interval of a 10-day washout period for the nutrigenomic study. This phase was performed at the Laboratory of Cellular Immunology, Faculty of Medicine, University of Brasilia, Brasilia. The results of the data obtained in study phase 2 are reported in Section 3.4, Section 3.5 and Section 3.6. For each experimental period, the fasting volunteers consumed either 250 mL of placebo drink (PB) or of *P. setacea* (PS) at the moment of their arrival in the morning, and blood samples were collected 3 h later.

This study was performed with the approval of the National Health Research Ethics Committee (CONEP, Brasilia, Brazil), protocol number 36348114.3.0000.0030, and all the volunteers provided their written informed consent. Description of the study can be found on ensaiosclinicos.gov.br RBR-84z83n.

### 2.3. Blood Sampling and Treatment

From the blood sampled in study phase 1, serum and plasma fractions were prepared to quantify metabolic markers (including glucose, insulin, homeostasis model assessment of β-cell function (HOMA-BETA), homeostatic model assessment for insulin resistance (HOMA-IR), total cholesterol, high-density lipoprotein (HDL), low-density lipoproteins (LDL) and total triglycerides) and cytokines. The collection of biological samples and the biochemical analysis were conducted by Sabin laboratory on the same day. Blood sampling was also collected in heparin tubes and stored at −80 °C for the later quantification of cytokines. In study phase 2 (Laboratory of Cellular Immunology), blood samples were collected in heparin tubes for further nutrigenomics analysis on isolated PBMC. A total of 8 mL of venous blood was collected from volunteers using BD Vacutainer tubes (Becton Dickinson, Franklin Lakes, NJ, USA), and PBMCs were isolated. Briefly, the tubes were immediately centrifuged at room temperature for 20 min at 1500 × *g*. After centrifugation, the cell layer containing PBMCs was collected and washed twice with sterile phosphate-buffered saline (PBS) with centrifugation at 300× *g* for 10 min after each washing step. The cell pellet obtained was immediately frozen at −80 °C and kept at this temperature until use.

### 2.4. Biochemical Parameters and Cytokines Analysis

The biochemical analyzes were conducted by Sabin Laboratory, Brasilia. To evaluate glucose, the hexokinase method was used; as for insulin, the insulin chemiluminescent immunoassay was applied, then HOMA BETA and HOMA IR were calculated. Total cholesterol was verified by means of the Allain’s method of esterase/oxidase [31], HDL by using the direct method, LDL through the Martin–Hopkins’s calculation, and total triglycerides by means of the oxidase/peroxidase method.

To quantify the circulating cytokines, serum samples were used to measure interleukin-2 (IL-2), interleukin-4 (IL-4), interleukin-6 (IL-6), interleukin-10 (IL-10), tumor necrosis factor (TNF), interferon-γ (INF-γ), and interleukin-17 (IL-17) protein levels by using the CBA Human T-cell TH1/TH2/TH17 Cytokine kit (Becton Dickinson, Franklin Lakes, NJ, USA). This method used bead array technology to simultaneously detect multiple cytokine proteins in the samples by flow cytometry. All the analyses were executed according to the manufacturer’s guidelines. Shortly, cytokine capture beads were mixed with the plasma samples and incubated with phycoerythrin (PE)-conjugated detection antibodies to form sandwich complexes. The FCAP Array software was used to generate results in graphical and tubular format.

### 2.5. Total RNA Extraction

Total RNA extraction from PBMC was performed by using RNeasy Mini Kit, as recommended by the manufacturer (Qiagen, Hilden, Germany). The RNA quality was checked by means of 1% agarose gel electrophoresis, whereas the quantity was checked through absorbencies at 260 and 280 nm on NanoDrop ND-1000 spectrophotometer (Thermo Scientific, Wilmington, DE, USA). The RNA samples were stored at −80 °C until use.

### 2.6. Microarray Analyses and Bioinformatic Analysis

Total RNA (50 ng per sample) was amplified and fluorescently labeled to produce Cy5 or Cy3 complementary RNA (cRNA) by using the Low Input Quick Amp Labeling Two-Color Kit (Agilent, Santa Clara, CA, USA) in the presence of a two-color spike-in control, as recommended by the manufacturer. After purification, 825 ng of labeled cRNA was hybridized onto G4845A Human GE 4x44K v2 microarray (Agilent, Santa Clara, CA, USA) according to the manufacturer’s instructions. The G4845A Human GE 4x44K v2 microarray contains 27,958 Entrez Gene RNA sequences. After hybridization, an Agilent G2505 scanner (Agilent, Santa Clara, CA, USA) was used to scan microarrays. The hybridization data were extracted by means of the Feature Extraction software version 11.0 and analyzed through the GeneSpring GX software version 14.5 (Agilent Technologies, Santa Clara, CA, USA). Data were normalized using 50th percentile shift and analyzed with moderated *t*-tests corrected by Westfall–Young permutation with corrected *p*-value cut-off set to 0.05. All transcripts presenting *p* < 0.05 were considered differently expressed.

### 2.7. Bioinformatic Analyzes

For biological interpretation of the differentially expressed genes, we first performed Gene Ontology (GO) analyses using DAVID (Database for Annotation, Visualization and Integrated Discovery v6.7). The GO results were grouped on the basis of their functionality by using the online REVIGO software. The partial least squares discriminant analysis (PLSDA) plot was obtained through MetaboAnalyst (https://www.metaboanalyst.ca). Gene networks were built with a data-mining approach using the Metacore software, and gene pathway analyses of the Kyoto Encyclopedia of Genes and Genomes (KEGG) and BioCarta databases were conducted by using the Genetrial2 online tool.

### 2.8. Statistical Analyses

The data obtained were previously analyzed for normality through D’Agostino’s and Pearson’s tests. The outlier values were calculated by means of the Tukey test and excluded from the analyses only when interfering with normality values. For two independent groups, paired Student’s *t*-test was applied to samples that had a normal distribution, and for those without normal distribution, Wilcoxon’s *t*-student test was applied. Descriptive values were expressed as mean ±SD corrected. The differences between the variables compared were considered statistically significant when the bi-tailed probability of their occurrence due to chance (type I error) was less than 5% (*p* < 0.05). Analyses and graphs were performed by using the GraphPad Prism 7 software for Mac (GraphPad Software, San Diego, CA, USA).

## 3. Results

### 3.1. Volunteers’ Baseline Characteristics

The baseline characteristics of the volunteers are summarized in Table 1. The subjects enrolled were men with a mean age of 48.66 ± 6.82 years that were overweight or slightly obese (BMI ranging from 25.00 to 30.80) with a mean waist circumference of 96.83 ± 6.49 cm. The subjects ranged from normal to slightly hyperglycemic (*n* = 1, 103 mg/dL), as well as from normal to mildly hyperlipidemic (*n* = 2), as shown by the values for plasma total triglycerides (Table 1). All the other parameters were within normal range. Two of the 12 volunteers presented three to four factors that may define them as having a metabolic syndrome (waist circumference ≥ 90cm; serum triglycerides ≥ 150 mg/dL mmol/l; HDL cholesterol < 40 mg/dL; and fasting plasma glucose (FPG) ≥ 100 mg/dL). Statistical tests without these volunteers were therefore remade and the statistical significances did not change.

### 3.2. Effect of Passiflora setacea Juice on Glucose and Lipid Metabolism (Phase 1)

Glucose, insulin, HOMA IR, triglycerides and HDL in plasma were analyzed before (T0) and 3 h after (T3) the intake of placebo and PS juice. The data show that insulin and HOMA IR levels decreased statistically 3 h after PS juice intake (*p* = 0.0068 and *p* = 0.001, respectively), whereas no significant change was observed after the placebo intake (Figure 1A). The plasma glucose concentrations decreased in a similar way after the intake of the two drinks. The high-density lipoprotein (HDL) level increased significantly after PS juice consumption (*p* = 0.0280), whereas no change was observed after PB drink (*p* = 0.3541), as seen in Figure 1B. No effects of PS or PB were detected on total and LDL cholesterol levels.

### 3.3. Effect of Passiflora setacea Juice Intake on Cytokine Serum Levels (Phase 1)

We determined the effect of PS juice intake on cytokine serum levels. Data showed that the IL-17A level did not increase after 3 h of PS juice consumption (*p* = 0.2962); however, it increased after placebo consumption (*p* = 0.0124) (Figure 2). We also observed that TNF-α presented a similar but not significant pattern as IL-17A, that is, its level tended to increase after PB drink (*p* = 0.0645), whereas it remained unchanged after PS juice (*p* = 0.5489) (Figure 2). There were no statistical changes in the other cytokine measures of IL-2, IL-4, IL-6, IL-10, and INF-γ (Figure 2).

### 3.4. Passiflora setacea Modulated Gene Expression in Circulating Cells (Phase 2)

Following RNA extraction and quality control of both RNA and microarray hybridization, we obtained good quality RNA from 8 out of 12 volunteers. To access the nutrigenomic effect of an acute intake of PS juice in PBMCs, we performed a pangenomic gene expression analysis 3 h after PS juice and PB drink consumption for the eight volunteers. Comparison of global gene expression profiles obtained for the volunteers by using PLSDA showed the separation of profiles between the two groups, suggesting different gene expression profiles between the volunteers that consumed PS and the volunteers that consumed PB (Figure 3). This observation suggests differential modulation in expression of genes after an acute intake of PS juice compared to the control drink.

Following this observation, we performed a statistical analysis to identify which genes had their expression altered after the consumption of PS compared to PB. Statistical analyses identified 1327 genes presenting changes in their expression after PS consumption. Among them, most genes were identified as having their expression down-regulated with the average fold-change for up-regulated genes being 2.48 and for down-regulated genes being −2.15. Among the genes showing the highest differential modifications were *TMEM151A*, *MLPH*, *MYH2*, *SERPINA9,* or *FNA21*.

For the biological interpretation of the differentially expressed genes, we first performed Gene Ontology (GO) analyses using DAVID database, and the GO were then clustered into function groups by using the online REVIGO software. This showed that differentially expressed genes are involved in various biological processes such as calcium ion transmembrane transport (potassium ion transport and phospholipid efflux), cell differentiation (extracelular matrix organization and histone lysine methylation), G-protein-coupled receptor signaling pathway (chemical synaptic transmission and neuropeptide signaling pathway), cell adhesion, and transcription from RNA polymerase promoter (Appendix A). This analysis revealed that the consumption of PS juice modulated the expression of genes presenting different biological functions.

To deepen the identification of the functions and cellular processes potentially affected by the consumption of PS juice, we performed gene network analyses of the differentially expressed genes. Gene networks, built through data-mining using the Metacore software, suggested, as did the GO analyses, that the consumption of PS juice changed the expression of genes involved in cellular function. Among the most over-represented networks were those involved in calcium and potassium transport, cell adhesion and cell–matrix interactions, neurogenesis, or transmission of nerve impulse (Figure 4). The genes identified in these networks were *NMDA* receptor, *matrix metalloproteinase (MMP)-7*, *ADAMTS9*, *mGluR*, *CaMKII* alpha, *CACNA1C*, and *SLC24A2*. These genes decode proteins involved in vascular tissue damage, in the reduction of insulin sensitivity and secretion, and in neuropsychiatric disorders and neuron excitability.

Besides the network analyses, we also performed gene pathway analyses employing the use of the Genetrial2 online tool by searching the KEGG and BioCarta databases. As shown in Figure 5, the differentially expressed genes identified are involved in cellular pathways including inflammation, metabolism, cell signaling, and neurofunction-related processes. Regarding the pathway related to inflammation, we identified a cytokine–cytokine receptor, cell adhesion molecules, and chemokine signaling pathways, which include genes such as *TNFSF18*, *IL36A*, *JAM2*, *ADCY8*, or *CCL16*. In pathways related to cellular metabolism, we identified circadian entrainment, insulin secretion, and *P13K-Akt* signaling pathways, which include genes such as *GRIN2A*, *PRKG1*, *CACNA1D*, *GLP1R*, *G6PC2*, and *LAMA1*. Calcium and adrenergic signaling pathways were also identified, containing *PTGER3*, *ADCY8*, and *CACNA1D* genes. Several pathways related to neurofunction were also identified such as glutamatergic synapse, neuroactive ligand-receptor interaction, and *MAPK* signaling pathways, in which genes such as *GABRG1*; *glutamate receptor*, *ionotropic*, *NMDA1* (*GRIN1*); *CACNA1D*; and *ADCY8* were mapped.

### 3.5. Protein–Protein Interaction (Phase 2)

Apart from the bioinformatics analyses on cellular networks and pathways of differentially expressed genes identified, we also searched for protein–protein interactions. We observed interactions among the genes whose expression were affected by the consumption of PS. By using the online String database, we identified 1013 nodes and 3066 edges with 6.05 nodes on average (Appendix A). Among them, 39 genes showed over 15 interactions with other proteins, making them important nodes in the protein–protein interactome. These 39 genes revealed over 700 interactions, which are a fifth of the total number verified. This suggests that changes in the expression of these genes can have an important impact on protein interactome and consequently on cell function. These genes are involved in the cellular pathways regulating processes such as cyclic adenosine monophosphate (cAMP) signaling pathway, nitric oxide signaling pathway, dopaminergic synapse, insulin, or PI3K-Akt signaling pathways. Proteins interacting with these 32 genes have been searched and 256 genes have been identified. Pathway analyses of these genes showed that they are involved in pathways such as the cAMP signaling pathway, PI3K-Akt signaling pathway, Ras/Rap1 (Ras-related protein 1) signaling, insulin secretion, cytokine–cytokine receptor interaction, chemokine signaling pathway, retrograde endocannabinoid signaling, or regulation of actin cytoskeleton.

### 3.6. Transcriptional and Post-Transcriptional Regulators of the Nutrigenomic Effect (Phase 2)

The bioinformatics analyses of gene expression data were further performed with the aim to identify potential transcription factors involved in the mediation of the PS juice nutrigenomic effect observed. The most significant transcription factors (Figure 6) were cAMP Responsive Element Binding Protein 1 (CREB1), nuclear factor-kappa B (Nf-kB), and specificity protein 1 (SP1). These transcription factors are involved in gluconeogenesis regulation, lipid metabolism, and insulin signaling pathways. They are also associated with vascular calcification, pathogenesis of type 2 diabetes (TD2), and diabetic cardiovascular disease. Other transcription factors are Proto-Oncogene C-Jun (c-jun), Signal Transducer And Activator of Transcription 3 (STAT3), and tumor protein P53 (p53).

Besides the transcriptional regulators potentially involved in the nutrigenomic effect observed, we also searched for potential post-transcriptional regulators of the gene expression, particularly microRNA. Using the online OmicsNet tool, we identified over 30 microRNAs (miRNAs) that could interact with differentially expressed genes and regulate their expression at the post-transcriptional level (Table 2). This suggests that PS juice consumption could potentially regulate the expression of microRNAs and consequently affect levels of mRNA of genes we identified as differentially expressed. Using the same online tool, we then performed integrated analyses of the identified differentially expressed genes, potential transcription factors, and potential microRNAs (Figure 7). We observed a network of interaction among these three levels of regulation of cell function, suggesting that PS juice consumption can significantly impact immune cells at the molecular level and consequently impact their functions.

## 4. Discussion

This is the first time that the effect of a Brazilian savannah fruit was described on IL-17A blood levels and on gene expression profile in humans. We found that the consumption of one serving of PS juice (similar composition of a whole fruit but without its seeds) statistically decreased the levels of insulin and HOMA IR while increasing HDL levels.

It is known that disorders in insulin metabolism and consequently in glucose metabolism result in oxidative stress and inflammation, which lead to micro- and macrovascular dysfunctions and to the further development of diabetes and cardiovascular diseases [5]. These complications are associated with endothelial dysfunction, pro-inflammatory cytokines, reactive oxygen species formation, and adhesion molecule production [32]. This process results in the increase in the adhesion of immune cells to endothelial cells, as well as in their transendothelial migration into the vascular wall, which are the initial steps to the development of atherosclerosis, the origin of all vascular-related diseases. Therefore, by changing the insulin, HOMA IR, and HDL levels, PS juice could exert anti-inflammatory and vasculo-protective properties and consequently prevent or delay the onset of associated diseases. This observation could be related to the results of other in vivo and in vitro studies that have shown the potential of dietary polyphenols on insulin response. For example, isoorientin, a flavonoid found in PS, has proven to revert insulin resistance in adipocytes by stimulating the proper phosphorylation of proteins in the insulin signaling pathway [33]. Another path of action may be explained by the inhibition of the key enzymes involved in starch digestion (alpha-amylase and alpha-glucosidase) by polyphenols [34]. The caffeic acid, a phenolic acid also present in PS has shown the capacity to inhibit these enzymes [35]. Polyphenols from water chestnut husk [36], green tea [37], and apple [38] have been reported to reduce serum insulin levels in normal mice. However, few studies have also shown the effect of whole food on human insulin metabolism. Nyambe-Silavwe and Williamson [39] reported the effect of dried fruits with green tea in the decrease of insulin serum levels in healthy volunteers.

We also observed an increase of blood HDL concentrations after PS juice intake that was not detected within the PB condition. Recent nutrition intervention studies on polyphenol-rich foods such as olive oil [40] or dark chocolate [41] have been shown to positively affect HDL levels in humans. One possible explanation is that these polyphenol-rich foods may induce changes in the biochemical properties of the lipoprotein that contribute to its main biological function, particularly in the enhancement of the cholesterol efflux capacity [42]. It can therefore be suggested that PS consumption modulates risk factors of cardiometabolic diseases.

This study also revealed that an acute intake of PS juice kept IL-17A at a basal level and indicated a tendency to decrease TNF-alpha levels (*p* = 0.0645) when compared to PB condition. The IL-17A is a pro-inflammatory cytokine that stimulates neutrophil inflammatory response [43] and the production of other pro-inflammatory cytokines such as TNF-alpha, IL-1B, and IL-6 [44], as well as the expression of adhesion molecules such as Intercellular Adhesion Molecule 1 (ICAM-1) [45]. Its activities are vastly increased due to synergy with TNF-alpha that promotes the induction of target genes involved in inflammatory processes [46]. Cyanidin, a key flavonoid present in red berries, has shown the capacity to reduce inflammation in mice through binding with the extracellular domain of IL-17RA and consequently disrupting the IL-17A/IL-17RA complex formation [47]. Few studies have provided evidence regarding the role of diet in modulating IL-17 levels in humans. Peluso et al. [6] observed a drop of this cytokine in the plasma of 14 overweight subjects after a pineapple, blackcurrant, and plum juice consumption. Taken together, this observation suggests that PS consumption could present anti-inflammatory effects during the post-prandial period.

In the present study, using microarray analysis, we showed that the consumption of one single cup of PS juice by volunteers significantly affected PBMC gene expression profiles. Our study is the first to show the effect of a *Passiflora* species on the modulation of gene expression in humans. Another study has shown the capacity of another species of *Passiflora* in modulating gene expression in mice. Toda et al. [48] demonstrated that the aerial parts of the *Passiflora incarnata* Linnaeus extract can modulate the expression of genes that may be involved in the prevention of obesity [49] and hyperglycemia [50]. Regarding the effects of BC found in PS on gene expression, it has been reported that isoorientin stimulated the transcription of genes encoding components of insulin signaling pathway in murine insulin-sensitive and insulin-resistant adipocytes [33]. Orientin from *Commelina communis* L. down-regulated the expression of peroxisome proliferator activated receptor (PPAR) and mRNA levels of genes involved in adipogenesis, lipogenesis, and triglyceride sysnthesis in vitro [51]. A plant extract rich in orientin, isoorientin, vitexin, and isovitexin has been shown to inhibit the mRNA levels of TNF-α also in vitro [52]. Therefore, our original study suggests that potential health benefits of PS could be related to its capacity to modulate the expression of genes in vivo in humans.

Bioinformatic analysis also revealed that PS consumption modulated the expression of a group of genes, 25, involved in the regulation of inflammation and immune response, particularly chemokine signaling pathway and cytokine–cytokine receptor interactions. Among these genes are CXCL17, IL36A, CCL16, CCL21, and IL-25. CCL16 is a pro-inflammatory chemokine that may be involved in the development of diseases such as irritable bowel syndrome [53]. Chemokines, a group of cytokines that attract and activate leucocytes into inflamed tissue, have been associated with the pathogenesis of a number of diseases, ranging from atherosclerosis to human immunodeficiency virus (HIV) infection [54], and CCL21 has been suggested as being involved in the pathogenesis of various inflammatory disorders including rheumatoid arthritis, inflammatory bowel diseases, and atherosclerosis [55]. Nutrigenomic analysis also identified several interleukins such as IL-36A that have pro-inflammatory properties and have been described as being involved in pulmonary inflammatory responses [56]. Several studies have shown that foods rich in BC or isolated BC such as hesperidin can regulate the expression of chemokines [7]. Taken together, the results suggest that the acute consumption of PS can present anti-inflammatory effects by modulating the expression of related genes.

This nutrigenomic study identified changes in the expression of genes involved in processes such as cell adhesion and cell–matrix interactions; chemokine signaling; insulin secretion; calcium and potassium transport; as well as inflammation, atherosclerosis development, and neurostimulation. Among them are genes encoding matrix metalloproteinases (MMPs), desintegrins, and metalloproteases (ADAMs), for which expression was identified as down-regulated. MMPs constitute a family of extracellular processing enzymes responsible for inflammation and acquired immunity [57]. Expression of genes encoding MMPs is frequently increased by cytokines, and reactive oxygen species are often involved in this mechanism [58]. MMP-7 over-expression regulates chemokine gradients that can lead to severe tissue damage through transepithelial influx of neutrophils [59]. In an in vitro and in vivo study, resveratrol reversed the injury of human epithelial cells and attenuated such injury in mice through the inhibition of MMP-7 expression [60]. Kinase Insert Domain Receptor (KDR) (or Vascular Endothelial Growth Factor Receptor-2, VEGFR2) is a key receptor that promotes Vascular Endothelial Growth Factor (VEGF) to form mitosis and generate vascularization. VEGF promotes proliferation and migration of cells and activates matrix matalloproteinase secretion [61], which can lead to exacerbation of tissue damage during inflammation. VEGF is highly expressed in tissues undergoing growth or remodeling in cancer and atherosclerosis [62]. KDR’s expression was down-regulated with PS juice consumption, as well as JAM-2’s (junctional adhesion molecule 2) gene involved in leukocyte recruitment and extravasation under inflammatory conditions [52]. Few studies have suggested the capacity of foods or BC to modulate the expression of these genes. A formulation of Chinese herbs was capable of downregulating the expression of KDR and VEGF in a mouse with hepatocellular carcinoma [63]. Monfoulet et al. [64] revealed the capacity of curcumin to reduce endothelial junctional permeability. Thus, the capacity of PS to down-regulate the expression of this gene suggests a potential lower interaction of immune cells with vascular endothelial cells, which represent the initial steps of atherosclerosis development. As atherosclerosis is associated with the genesis of others cardiovascular diseases, PS juice consumption may reveal interesting mechanisms underlying its potential vasculo-protective properties.

ADAMs constitute a family of proteases with cell adhesive potential [65] and other functions, including extracellular matrix (ECM) degradation, shedding of various cell surface proteins, and influence on cell signaling patterns [66]. ADAM12 is an active protease in ECM that causes changes in proliferation and differentiation of adipocyte maturation and also in the development of obesity induced by high-fat diet [67]. It has been shown to affect the insulin-like growth factor (IGF)/mTOR (mammalian target of rapamycin) and peroxisome proliferator-activated receptor gamma (PPARγ) signaling pathways, leading to increased lipid accumulation in mature adipocytes [68]. ADAMTS9 is a risk gene for type 2 diabetes development and its over-expression is associated with impaired insulin signaling in peripheral tissues and also with insulin resistance [69]. Its risk allele (rs4607103 C) has been demonstrated to decrease mitochondrial function and to alter glucose and lipid metabolism [70]. Another gene identified as differentially expressed after PS juice consumption is adenylate cyclase 8 (ADCY8), which is involved in insulin secretion and glucose homeostasis. Sung et al. [71], in a genome-wide association analysis, associated the ADCY8 gene with obesity and abdominal visceral fat depot. Considering all these previous factors, the gene expression profile obtained after an acute consumption of PS suggests a lower accumulation of lipids in PBMCs and a lower impairment in insulin signaling, presenting potential molecular targets of PS juice consumption and their potential health properties.

Besides the modulation of genes related to cardiometabolic regulations, our nutrigenomic analysis also identified the fact that several genes modulated by the acute consumption of PS were associated with neurofunction, such as CACNA1C, GRIN1, and G protein-coupled receptor 50 (GPR50). The immune-to-brain and brain-to-immune communication has been recently studied [72]. The central nervous system can communicate with peripheral monocytes, promoting gene expression modulation, particularly with regards to the NF-kB transcriptional control pathway [73]. The CACNA1C is a gene contributes to the etiology of psychiatric disorders and to phenotypes affected by those conditions such as memory and circadian rhythms [74]. Such symptoms are present in a proportion of the general population and are correlated with poorer cognitive performance and with adverse health outcomes [75] such as atherosclerosis [76]. The GRIN1 (glutamate receptor, ionotropic, NMDA1) gene plays an important role in excitatory neurotransmission, and the increase of its expression has been associated with anxiety in response to stress in mice [77]. Another gene whose expression has been modulated after PS juice consumption was GPR50, a gene involved in late-life depression in certain subgroups of depressed individuals [78]; its down-regulation is associated with torpor enhancement [79]. Acute consumption of PS juice decreased the expression of these genes, which suggests that PS consumption may affect torpor, a hypothesis that can explain PS’s popular name “sleep passionfruit”. These observations suggest the potential effect of PS consumption as an auxiliary treatment for cognitive functions and for psychiatric disorders.

The three major transcription factors whose activities might be affected by PS consumption and that are possibly involved in the nutrigenomic effect observed are CREB1, RELA Proto-Oncogene (RelA), and SP1. CREB1 regulates gluconeogenesis, lipid metabolism, and insulin signaling pathways, and the activity of its transcriptional promoter is associated with the pathogenesis of TD2 [80], adipogenesis [81], and major depressive disorder [82]. RelA is a sub-unit of NF-κB, a transcription factor critical for the expression of proinflammatory cytokines in human monocytes. Studies on the effect of bioactive compounds (BC) on this transcription factor showed that orientin, isoorientin, vitexin, and isovitexin exert suppressive action of these compounds upon NF-κB activation [24]. Elevated SP1 plays a pro-apoptotic effect and stimulates vascular calcification [83]. It has been observed that the BC (-)-epigallocatechin-3-gallate can modulate the activity of this transcription factor [84]. Therefore, the capacity of PS juice consumption to affect the activity of these transcription factors could present major regulatory mechanisms observed in nutrigenomic modifications underlying their health properties.

Literature lacks data about the effect of *Passiflora* on miRNA expression. However, it has been suggested that plant food bioactives can modulate the expression of miRNA [85,86]. Among the miRNA identified from bioinformatic analysis as potentially involved in the post-transcriptional regulation of identified differentially expressed genes by PS juice consumption, we can list miRNA-16, -26, and -124. miRNA-16 has shown to play a role in inflammation [87] and hypertension [88], whereas miRNA-124 mediates anti-inflammatory effects and is involved in neuroprotective mechanisms [89]. Therefore, by potentially modulating the expression of these miRNAs, PS consumption can regulate genes involved in the development or the prevention of cardiometabolic and neurological diseases.

## 5. Conclusions

This acute study has provided the first clinical data supporting an interest in PS pulp consumption for human health. The positive changes we observed in some inflammatory and metabolic biomarkers as well as in the PBMC gene expression profile after an acute intake of one serving of PS pulp by overweight, middle-age volunteers suggest potential anti-inflammatory and anti-diabetic effects. These results, originating from an acute study, cannot be directly extrapolated to chronic consumption of PS pulp. However, it opens interesting perspectives for future long-term clinical studies using PS products in order to better characterize their health properties and identify the compounds/components responsible for these biological effects. Furthermore, developing this research line is important to increase knowledge on the role of plant foods derived from Brazilian biodiversity in the prevention or delay of the onset of chronic diseases.

## Figures and Tables

**Figure 1 nutrients-12-01104-f001:**
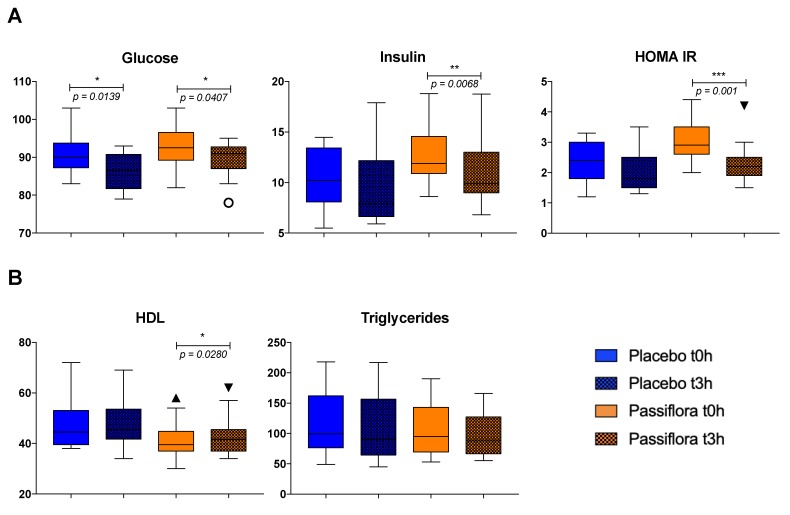
Acute effect of *Passiflora setacea* juice and placebo drink consumption on glucose (**A**) and lipid (**B**) metabolism markers in overweight volunteers (*n* = 12). Results analyzed by means of the non-parametric paired *t*-test (Wilcoxon’s test), medians, and SD. * *p* ≤ 0.05, ** *p* ≤ 0.01, *** *p* ≤ 0.001.

**Figure 2 nutrients-12-01104-f002:**
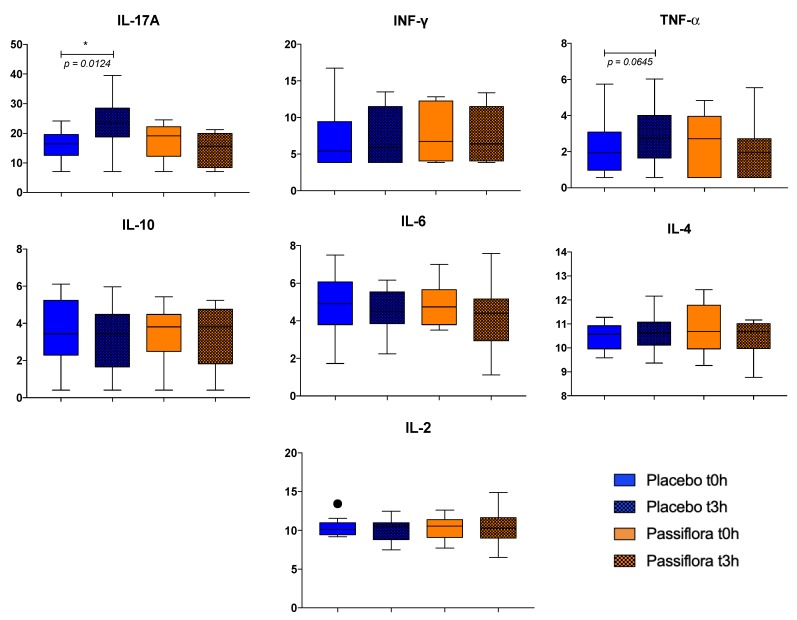
Acute effect of *Passiflora setacea* juice consumption on cytokine serum levels in overweight volunteers (*n* = 12). Results analyzed by means of the non-parametric paired *t*-test (Wilcoxon’s test). * *p* ≤ 0.05, • outlier tested by Tukey.

**Figure 3 nutrients-12-01104-f003:**
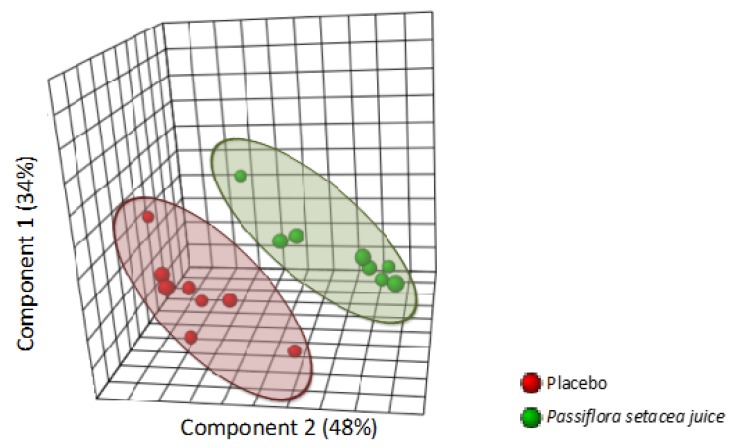
The comparison of the global gene expression profiles obtained for the volunteers using partial least squares discriminant analysis (PLSDA) shows the separation of profiles between the two groups, and suggests different gene expression profiles between the volunteers that consumed *Passiflora setacea* juice (PF) and the volunteers that consumed placebo (PB; placebo).

**Figure 4 nutrients-12-01104-f004:**
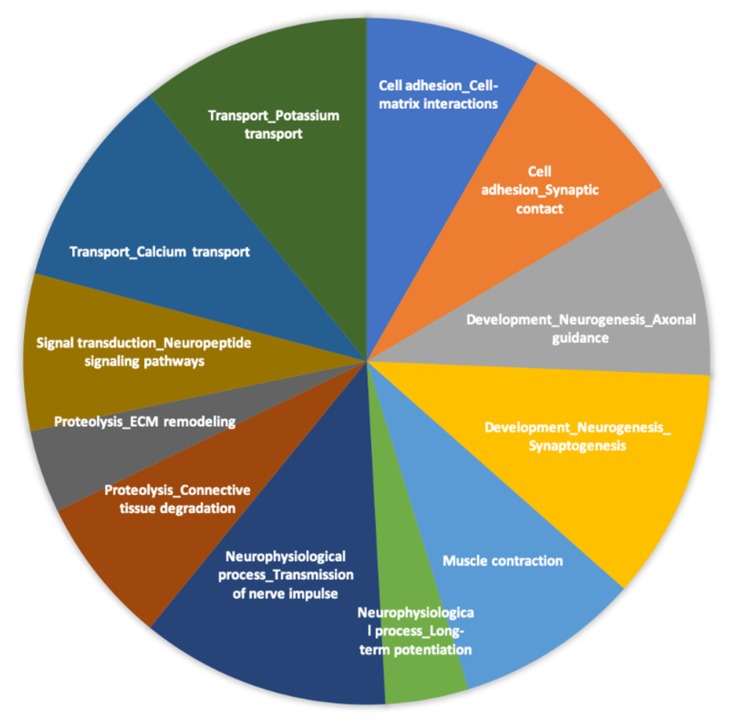
Networks enriched with differentially expressed genes in volunteers’ peripheral blood mononuclear cells in response to *Passiflora setacea* pulp consumption. Gene networks were identified using MecaCore software that uses text mining approach to build gene–gene interactions and identify their cellular functions.

**Figure 5 nutrients-12-01104-f005:**
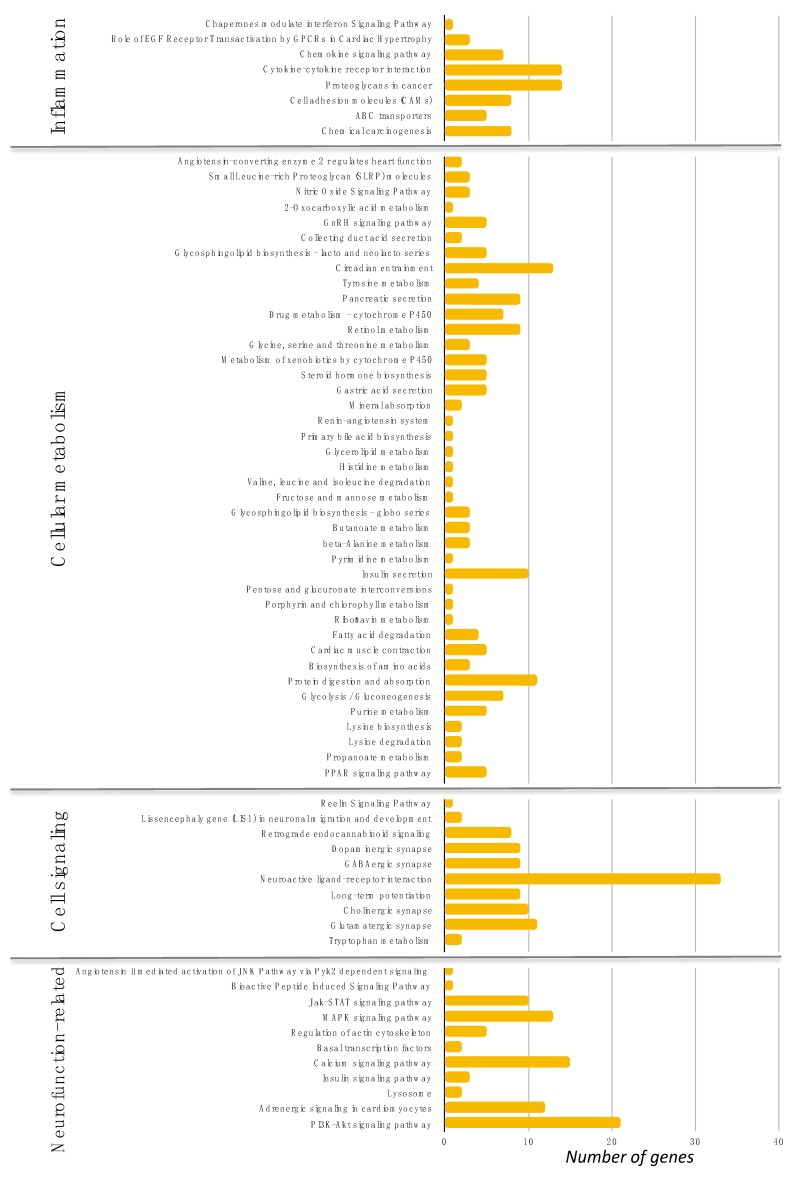
Significantly enriched pathways with differentially expressed genes in volunteers’ PBMC in response to PS pulp consumption. Pathways were identified using Genetrial2 online tool and KEGG database, and were grouped regarding their functions. *x*-axis represents the number of genes in each pathway.

**Figure 6 nutrients-12-01104-f006:**
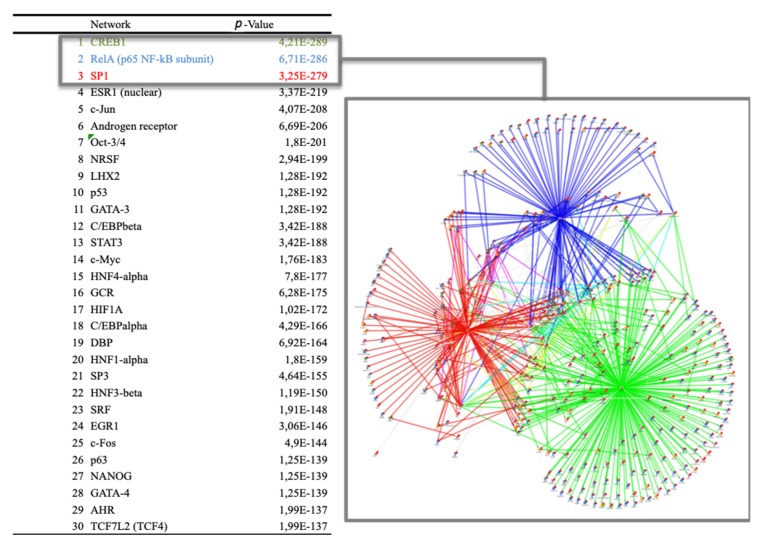
Bioinformatics analyses of potential transcription factors involved in the mediation of the PS juice’s nutrigenomic effect observed. Transcription factors were identified using MetaCore algorithm and the most significant transcription factors listed were CAMP Responsive Element Binding Protein 1 (CREB1), nuclear factor-kappa B (Nf-kB), and specificity protein 1 (SP1). On the right, the interactions of the networks with these three transcription factors identified are represented.

**Figure 7 nutrients-12-01104-f007:**
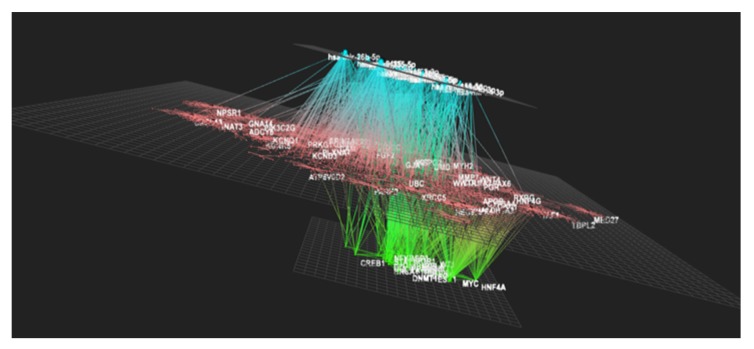
Bioinformatics analyses of potential miRNA involved in the mediation of the *Passiflora setacea* juice’s nutrigenomic effect observed. OmicsNet online tool was used to identify interactions between the differentially expressed genes identified (in pink) with the potential transcription factors identified (in green), and potential miRNAs involved (in blue).

**Table 1 nutrients-12-01104-t001:** Volunteers’ baseline characteristics (*n* = 12).

Parameter	Mean ± SD (Range)
Age, years	48.66 ± 6.82 (41–62)
BMI, kg/m^2^	28.18 ± 2.08 (25.0–30.8)
Waist circumference, cm	96.83 ± 6.49 (88–112)
Fasting glucose, mg/dL	90.83 ± 5.54 (83-103)
Basal insulin, μUI/mL	11.14 ± 3.58 (5.5–17.7)
HOMA IR	2.49 ± 0.77 (1.2–3.7)
HOMA BETA	152.8 ± 71.58 (73–304)
Triglycerides, mg/dL	116.58 ± 57.55 (49–218)
Total cholesterol, mg/dL	188.17 ± 33.40 (115–232)
HDL cholesterol, mg/dL	47.91 ± 10.67 (38–72)
LDL cholesterol, mg/dL	117.00 ± 27.09 (60–156)
Apoliprotein A, mg/dL	133.67 ± 20.95 (107–172)
Apoliprotein B, mg/dL	101.77 ± 24.80 (49–133)
Creatinin, mg/dL	0.97 ± 0.14 (0.7–1.2)
TGO, U/L	20.67 ± 2.87 (14–25)
TGP, U/L	25.41 ± 6.97 (11–35)

**Table 2 nutrients-12-01104-t002:** microRNA list.

hsa-mir-335-5p	MIMAT0000765	228
hsa-mir-26b-5p	MIMAT0000083	193
hsa-mir-16-5p	MIMAT0000069	161
hsa-mir-92a-3p	MIMAT0000092	161
hsa-mir-124-3p	MIMAT0000422	152
hsa-mir-155-5p	MIMAT0000093	116
hsa-mir-93-5p	MIMAT0000646	116
hsa-mir-1-3p	MIMAT0000416	114
hsa-mir-17-5p	MIMAT0000070	113
hsa-mir-615-3p	MIMAT0003283	112
hsa-let-7b-5p	MIMAT0000063	104
hsa-mir-106b-5p	MIMAT0000680	101
hsa-mir-20a-5p	MIMAT0000075	99
hsa-mir-218-5p	MIMAT0000275	99
hsa-mir-1-1	MI0000651	98
hsa-mir-484	MIMAT0002174	90
hsa-mir-193b-3p	MIMAT0002819	89
hsa-mir-15b-5p	MIMAT0000417	88
hsa-mir-15a-5p	MIMAT0000068	81
hsa-mir-20b-5p	MIMAT0000076	80
hsa-mir-21-5p	MIMAT0001413	80
hsa-mir-30a-5p	MIMAT0000087	79
hsa-mir-186-5p	MIMAT0000456	78
hsa-mir-519d-3p	MIMAT0002853	77
hsa-mir-24-3p	MIMAT0000080	76
hsa-mir-320a	MIMAT0000510	75
hsa-mir-8485	MIMAT0033692	75
hsa-mir-192-5p	MIMAT0000222	74
hsa-mir-195-5p	MIMAT0000461	73

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
