# Peer review of "Acute Effects of the Consumption of Passiflora setacea Juice on Metabolic Risk Factors and Gene Expression Profile in Humans"

_nutrients, 2020, doi:10.3390/nu12041104_

Round 1
Reviewer 1 Report
All my comments have been addressed.
Author Response
The reviewer said that all his comments have been adressed.
Reviewer 2 Report
The manuscript “Acute effects of the consumption of Passiflora setacea juice on metabolic risk factors and gene expression profile in humans” by de Araújo Esteves Duarte et al. studied the impact of consuming a juice from P. Setacea on metabolic risk factors in humans, through via the evaluation of biochemical and immunological markers and gene expression.
This manuscript is now better structured and written. The experimental design proposed in the paper allows the authors to respond to most of their objectives. Following the recommendation of both reviewers, the authors have been able to increase the quality of their manuscript in comparison with the first version.
Nevertheless, the authors should make minor changes before getting their manuscript accepted for publication. The comments from the reviewer are listed below:
- The authors should transform Figure 4 into a 2D figure. Avoid the use of 3D.
- As previously stated, due to the complexity of figure analysis, the referee recommends including a self-explanatory caption in figures 4, 5, 6, and 7. Even if it could be quite repetitive, the fluidity of the manuscript will be enhanced.
- Create a separate conclusion section.
Author Response
- The authors should transform Figure 4 into a 2D figure. Avoid the use of 3D.
We’ve modified the Figure 4 as suggested by the reviewer. Please see page 9. We’ve also done this modification in the file “Figures, Graphics, Images”.
- As previously stated, due to the complexity of figure analysis, the referee recommends including a self-explanatory caption in figures 4, 5, 6 and 7. Even if it could be quite repetitive, the fluidity of the manuscript will be enhanced.
We’ve modified the legends of figures 4, 5, 6 and 7 as suggested by the reviewer. Please see lines 278, 279, 294, 295, 322-324, 339-341. We’ve also done this modification in the file “Figures, Graphics, Image”.
- Create a separate conclusion section.
A separate conclusion section was done. Please see line 495.

This manuscript is a resubmission of an earlier submission. The following is a list of the peer review reports and author responses from that submission.
Round 1
Reviewer 1 Report
The manuscript “Acute consumption of Passiflora setacea juice improves risk factors of cardiometabolic diseases and modulates gene expression profile in humans” by de Araújo Esteves Duarte et al. studies the effects of acute intake of a juice from P. Setacea in humans, through the study of biochemical and immunological markers and gene expression.
In general, the manuscript presents several errors, which should have been improved/resolved before submission.
The referee recommends the following Author’s instructions for the following submissions. The manuscript should have presented Nutrients’ style. The authors should carefully revise both English and reference style.
The manuscript presents results with not enough robustness. Most biochemical markers present no changes after PS consumption. Inflammatory markers showed no changes either. However, the manuscript’s title says that PS improves risk factors of cardiometabolic diseases.
Regarding gene expression and bioinformatics analysis, the authors found a differential gene expression between the two study groups; however, the conclusions obtained using bioinformatics tools seem to be entirely hypothetical. There is no cohesion among the different analyses. Furthermore, only data from 8 volunteer was able to be analyzed, diminishing the sample size considerably.
The quality of the presentation of the results is poor. Figures are complicated to read and understand without a needed explanatory caption.
As it is, this manuscript is not ready to be published in Nutrients. The authors should perform some experiments and rewrite some parts of the manuscript to have their paper reconsidered for publication.
Reviewer 2 Report
The role of plant foods and their role in the prevention or delay of the onset of chronic disease is a topic worth investigating. This study provides evidence for acute positive effects of the administration of one serving of Passiflora setacea pulp on metabolic factors and altered gene regulation in PBMC of overweight (and some abdominally obese) middle aged men.
The authors’ efforts are appreciated.
The only and main major concern I have is on the selection of the study population.
1.1) I assume the authors chose to include overweight subjects in order to maximize the chance to observe differences between PB and PS conditions. The methods section 2.2, row 101, states as inclusion criteria a waist circumference ³ 102 cm, but in Table 1, the reported range of waist circumference is 82-107. How do the authors comment this discrepancy?
1.2) In the discussion, row 387, the authors define the overweight subjects “healthy”.
According to the IDF definition of Metabolic Syndrome (criteria for men):
Central obesity can be defined by waist circumference ≥94 cm in white Caucasian or Afro-Caribbean men (≥92 cm in a study of a Brazilian cohort, Cardinal TR et al. 2018) plus two of the following four factors:
Serum triglycerides ≥ 150 mg/dL mmol/l or specific treatment for this lipid abnormality. HDL cholesterol < 40 mg/dL in men or specific treatment for this lipid abnormality. Systolic blood pressure ≥ 130 or diastolic BP ≥ 85 mm Hg or treatment of previously diagnosed hypertension. Fasting plasma glucose (FPG) ≥ 100 mg/dL, or previously diagnosed type 2 diabetes.
This study population had a WC (mean ± SD) = 92 ± 6 (82-107), meaning that a some if not about half of the subjects fall in the category of abdominally obese subjects. It can be argued by some that also abdominally obese subjects could be “metabolically healthy”, but at least two of the subjects were hyperlipidemic (was the cut-off 150 mg/dL used?) and one was indeed only slightly hyperglycemic but yet above the cut-off. I wonder:
Was one subject falling in the IDF metabolic syndrome definition? Was the subjects’ blood pressure measured?
Minor comments.
2.1)Row 90, First occurrence of “FW”. Abbreviation not explained. Is it allowed by the journal?
2.2)Row 92, First occurrence of “DW”. Abbreviation not explained. Is it allowed by the journal?
2.3)Row 182, First occurrence of “David”. It is preferable to write as “DAVID”, please check following occurrences in the text.
2.4)Row 184, First occurrence of “PLSDA”. Abbreviation not explained. It could be written as “Partial Least Squares-Discriminant Analysis”
2.5)In the result section for better reading it should be clear if the results described are obtained by Phase 1 data or Phase 2 data.
Row 121. At the end of the sentence is a good place to add for example “Results of the data obtained in Phase 1 are reported in sections 3.2 and 3.3”. Row 128. After “Brasilia.” “Results of the data obtained in Phase 2 are reported in sections 3.4, 3.5 and 3.6”.
alternatively
in the title of the results section “3.2 Effects …. Lipid Metabolism (Phase 1)”
Same for the other sections
2.6) In row 201 the mean and and SD of age and in row 202 of waist circumference are not rounded properly in Table 1.
2.7) In row 210 p=0.0001 is different from p=0.001 in Figure 1 A.
2.8) Figure 1 and 2, I recommend use a different shade of blue color between PB t0 and t3 to be consistent with the style for PS t0 and t3.
2.9) In row 223 p=0.0251 is different from the p value reported in Figure 2 (p=0.0124)
2.10) I suggest adding “a” to TNF in the Figure 2.
2.11) Figure 3
- legend (in the figure). I suggest replacing “cont” with “control” or “PB”.
-Component 2 (48) is missing “%”
2.12) In row 247, the sentence starting with “Among them, most…”. I recommend replacing “most” with a numerical value (% of 1327 genes).
2.13) Figure 4, 5,6,7,9 appear to be of poor quality in terms of resolution and it is difficult to read the text.
- Fig 4. Maybe removing the bold from the font could improve readability if the resolution cannot be increased.
-Fig 4 and 5. As additional suggestion, if some of the groups mentioned include only down-regulated or up-regulated genes, it would be useful to introduce either a text color code for up-regulated only, down-regulated only or mixed, or adding some up , down or up and down arrows close to the names.
-Fig 6 legend. It could be added at the beginning “Number of hits in pathways…” (Optional)
-Fig 7 is not clear and not informative in this state, colors are not explained. I could recommend to cite a link and make it available online with higher resolution.
-Fig 9 legend, “ofthe”. Please add space.
2.14) In row 344, the data of the study do not show that PS juice statistically decreases levels of IL-17A. It should be removed “and IL-17A,”
2.15) In row 387, “aPassiflora’s”. Please add space.
2.16) In row 460, “T2DM” is preferred to “TD2”.